# Experimental Investigation of Fatigue Capacity of Bending-Anchored CFRP Cables

**DOI:** 10.3390/polym15112483

**Published:** 2023-05-27

**Authors:** Jingyu Wu, Yongquan Zhu, Chenggao Li

**Affiliations:** 1College of Civil Engineering, Shandong Jianzhu University, Jinan 250101, China; wujingyu1985@163.com; 2Key Laboratory of Building Structural Retrofitting and Underground Space Engineering, Shandong Jianzhu University, Ministry of Education, Jinan 250101, China; 3Liuzhou OVM Machinery Co., Ltd., Liuzhou 545006, China; yongquanzhu_ovm@163.com; 4Key Laboratory of Smart Prevention and Mitigation of Civil Engineering Disasters of the Ministry of Industry and Information Technology, Harbin Institute of Technology, Ministry of Education, Harbin 150090, China; 5Key Laboratory of Structures Dynamic Behavior and Control, Harbin Institute of Technology, Ministry of Education, Harbin 150090, China; 6School of Civil Engineering, Harbin Institute of Technology, Harbin 150090, China

**Keywords:** fatigue-bearing capacity, bending anchoring system, premature fiber breakage, acoustic emission

## Abstract

In this study, the variation of fatigue stiffness, fatigue life, and residual strength, as well as the macroscopic damage initiation, expansion, and fracture of CFRP (carbon fiber reinforced polymer) rods in bending-anchored CFRP cable, were investigated experimentally to verify the anchoring performance of the bending anchoring system and evaluate the additional shear effect caused by bending anchoring. Additionally, the acoustic emission technique was used to monitor the progression of critical microscopic damage to CFRP rods in a bending anchoring system, which is closely related to the compression-shear fracture of CFRP rods within the anchor. The experimental results indicate that after the fatigue cycles of two million, the residual strength retention rate of CFRP rod was as high as 95.1% and 76.7% under the stress amplitudes of 500 MPa and 600 MPa, indicating good fatigue resistance. Moreover, the bending-anchored CFRP cable could withstand 2 million cycles of fatigue loading with a maximum stress of 0.4 *σ*_ult_ and an amplitude of 500 MPa without obvious fatigue damage. Moreover, under more severe fatigue-loading conditions, it can be found that fiber splitting in CFRP rods in the free section of cable and compression-shear fracture of CFRP rods are the predominant macroscopic damage modes, and the spatial distribution of macroscopic fatigue damage of CFRP rods reveals that the additional shear effect has become the determining factor in the fatigue resistance of the cable. This study demonstrates the good fatigue-bearing capacity of CFRP cable with a bending anchoring system, and the findings can be used for the optimization of the bending anchoring system to further enhance its fatigue resistance, which further promotes the application and development of CFRP cable and bending anchoring system in bridge structures.

## 1. Introduction

The service life of cable components in infrastructure (such as suspenders in arch bridges and ground anchors in rock reinforcement) becomes a critical issue that determines its safety and cost when exposed to severe engineering environments. Since the 1990s, the unidirectional carbon fiber-reinforced polymer (CFRP) rod has emerged as the most promising alternative part to steel wire due to the CFRP’s superior tensile properties and corrosion resistance [1,2,3]. However, the high cost of CFRP restricts its large-scale applications. In recent years, the rapid development of CFRP mixed with lower-cost fiber [4] has reduced the cost significantly, and its lower life-cycle cost in the long-span cable-stayed bridge [5] further promoted its application in large-tonnage cable components. Over the past three decades, the durability of CFRP cables has been proven in several practical applications, such as the ground anchor in Aizhai Bridge [6] and the stay cable in Stock Bridge [7].

As a result of the anisotropic and weak-shear properties of CFRP rods, however, the anchoring performance has a significant effect on the ultimate bearing capacity (UBC) and fatigue resistance of CFRP cable [6,7,8,9,10,11]. The anchoring effect causes premature fiber breakage due to lateral shear stress or local fiber bending induced by adhesive and, in the worst-case scenario, leads to compression-shear fracture of the CFRP rod in the anchoring section. Consequently, over the past 20 years, numerous novel anchors have been proposed, and substantial experimental and numerical studies have been conducted to reduce the damage concentration of CFRP rods in the anchoring system. Zhang et al. [6] and Mei et al. [7] proposed a straight and a straight-inner cone bonding anchor, respectively, to diminish the compression stress on CFRP rods by optimizing the inner structure of the anchoring cup; Meier et al. [8] and Wang et al. [9,10] adopted gradient stiffness adhesives and continuous-fiber-reinforced load transfer component (LTC) to reduce the compression stress on CFRP rods near the loaded end of the anchoring section. The aforementioned studies reveal the damage mechanism of CFRP rods due to anchoring, and the findings become fundamental for the applications of CFRP cables.

In recent years, with the development and implementation of large-tonnage CFRP cables, the increased number of CFRP rods in the anchoring system has necessitated a rational and efficient spatial arrangement within a restricted volume of the anchoring system. Therefore, the arrangement of CFRP rods in the anchoring system has become important for the optimization of the anchor systems. The arrangements of CFRP rods in the anchoring system reported in the literature can be classified as either parallel or diverged. Parallel arrangement [7,8,9,10,12,13,14,15] favors the anisotropic nature of CFRP rods, but the interspaces between the CFRP rods (i.e., for the contact between the CFRP rods and the adhesive) increase the diameter of the CFRP rod bundle in the free section of the cable. In addition, the narrow interspace among CFRP rods in the anchoring system presents a higher requirement for the filling density of the adhesive [7], which usually affects the stress uniformity of the CFRP rod bundle. The divergent arrangement adopted for the first time by Zhang et al. [6] mitigates these problems substantially. In this arrangement of CFRP rods, a positioning plate is used to bend the rods prior to filling the potting compound, and an inner cone-shaped anchoring cup is incorporated to provide a larger interspace between the CFRP rods to enhance bonding and to reduce the nominal diameter of the CFRP rod bundle. Notably, the bending anchoring mode causes additional shear on CFRP rods within the anchoring section. This effect may aggregate premature fiber failure, resulting in compression-shear fracture of CFRP rods within the anchoring system. Consequently, the bending extent of a CFRP rod (closely associated with the level of additional shear) has been studied [6,16,17,18], and the maximum allowable bending extent has been determined based on the tensile test results of a CFRP rod anchored in the bending mode, with strength loss limited to 5% to ensure anchoring performance [18].

Up to date, for the generality and complexity of the fatigue-damage models for the unidirectional FRP materials subjected to multiaxial loading [19,20,21,22], most current studies on the optimization of the anchoring system were conducted based on stress analysis and cannot assess or predict the degree of damage to CFRP rods in the anchoring section under specific fatigue loading. Consequently, experimental studies remain an indispensable means to evaluate the performance of the anchoring system. In experimental investigations of the anchoring system for CFRP cable, the ultimate bearing capacity and fatigue resistance, which are related to the reliability and service life of CFRP cable, are among the most important factors in the development of the anchoring system. In contrast to UBC testing of CFRP cables, experimental research on the fatigue resistance of CFRP cables requires a significant amount of time and money. Consequently, most CFRP cable fatigue tests are based on practical engineering and follow the testing standards established for steel cables [7,12,13,14,16]. Due to the excellent fatigue resistance of CFRP rods, almost all tested CFRP cables reported in the literature do not exhibit any symptoms of fatigue fracture resulting in the fatigue resistance capacity of these cables, which has mainly been evaluated on the basis of UBC post-fatigue loading. Therefore, current fatigue tests conducted on CFRP cables have shown that the fatigue-bearing capacity of CFRP cables employed in the currently developed anchoring system is far from being attained, and further investigation is required on the potential of CFRP cables to be used in environments with high levels of fatigue. However, very few reports on the fatigue-bearing capacity of cables are available in the literature. Li et al. [15] investigated the fatigue-damage progress of CFRP cable with fatigue stress ranging from 30 N to 0.25 times of UBC force and divided the fatigue-damage progress into three stages according to the characteristics of multi-mode microscopic damage evolution in each stage. Feng et al. [14] have experimentally investigated the fatigue-bearing capacity of CFRP cable with fixed maximal stress (0.45 fu) to investigate the fracture modes and fatigue life of CFRP cable subjected to different stress amplitudes. Notably, all of the experimental investigations on the fatigue-bearing capacity of CFRP cables were conducted on CFRP cables with parallel arrangements of CFRP rods in the anchoring system. The characteristics of fatigue-damage mode, progression, and fracture in bending-anchored CFRP cables subjected to severe fatigue loading are still unknown. Thus, a systematic study on the fatigue-bearing capacity of bending-anchored CFRP cables is necessary to further expand its applications.

Experimental investigations of the fatigue-bearing capacity of CFRP cables should be conducted under reasonable and typical fatigue-loading conditions in order to assess the damage caused by the anchoring system to CFRP rods. Thus, the fatigue-loading conditions applied to CFRP cable have a strong correlation with the fatigue properties of CFRP rods. Consequently, the fatigue properties of a CFRP rod under axial fatigue loading are essential to assessing the performance of the anchoring system used in CFRP cable. Experimental studies on the fatigue properties of CFRP rods have revealed that the anchoring system used for the fatigue test of a CFRP rod must be well-designed to prevent slippage and severe premature fiber failure within the anchoring section. Xie et al. [23] investigated slippage and the variation of bonding interfacial temperature in a bonded anchorage system during the fatigue-loading process, revealing their impact on the ultimate anchoring capacity. Song et al. [24] proposed a novel wedge-type anchorage system that could withstand fatigue stress amplitudes from 200 MPa to 800 MPa and maximal stresses from 0.37 to 1.0 fu without slippage or failure in the anchoring section. In general, the anchoring system utilized for a CFRP rod in a fatigue test must ensure that the fatigue damage propagates in the mature mode. On the other hand, experimental investigation of the fatigue-bearing capacity of CFRP cables involves macroscopic and microscopic fatigue damage caused to CFRP rods, especially the damage concentration in the anchoring section, which cannot be observed during fatigue loading and is closely associated with the premature fracture of CFRP rods inside the anchoring system. By capturing the stress wave emitted from a damaged material in real-time, acoustic emission (AE) technology is widely used for monitoring the microscopic damage caused to FRP materials [25,26,27]. AE monitoring studies on FRP materials have shown that an AE signal carries information related to damage, which enables identifying multi-mode damage patterns, e.g., matrix cracking, fiber-matrix debonding, and fiber breakage in FRP materials. Gutkin et al. [25] extracted the inherent relationship between the peak frequency of an AE signal and the FRP material damage patterns in material-scaled AE monitoring using AE frequency analysis. Therefore, the degree of damage in FRP material could be evaluated by the combination of AE technology and damage models. Several studies on AE damage monitoring in CFRP cables have been reported in the literature [12,15,17]. Based on the time-domain distribution of AE signals and the accumulated AE energy release, Rizzo et al. [12] and Li et al. [15] employed AE monitoring to assess the progression of fatigue damage in CFRP cables. It is worth noting that, in large-scale monitoring, the AE frequency analysis used to recognize multi-mode damage patterns in FRP material is limited by the attenuation of stress waves during their propagation, making the AE frequency characteristics distance-dependent [28]. Wu et al. [17] used cluster-based pattern recognition in UBC tests to analyze complex AE signals obtained from the anchoring section of CFRP cable to assess the damage caused to CFRP rods and the adhesive inside the anchoring cup. As a result, AE monitoring has become an effective method to trace the propagation of multi-mode damage caused to CFRP rods within the anchoring system.

To promote the application and development of CFRP cable with bending anchoring system in bridge engineering structures, the evaluation and analysis of fatigue capacity and damage mechanism from microscopic and macroscopic scales of CFRP cable subjected to severe fatigue-loading environments are urgent. In this study, the fatigue-bearing capacity of bending-anchored CFRP cables under several specific fatigue-loading conditions was experimentally evaluated, including the variation of fatigue stiffness, fatigue life, residual strength, microscopic and macroscopic damage initialization, expansion, and fracture of CFRP rods in the cable. Initially, the fatigue properties of CFRP rods under axial-fatigue loading are experimentally examined to provide a foundation for evaluating the effect of bending anchoring on the fatigue resistance of CFRP cable. Next, CFRP cables employed in a bending anchoring system are tested under specific fatigue-loading conditions to investigate the characteristics of macroscopic damage progression. Furthermore, AE technology is employed to monitor the multi-mode damage progression in CFRP rods/cables caused by fatigue loading. The experimental results reveal the microscopic and macroscopic damage progression in CFRP rods, and the findings of this work can be applied to the further optimization of the bending anchoring system that could be used in environments with more severe fatigue loading.

## 2. Materials and Methods

Experiments were conducted to determine the fatigue properties of CFRP rods under axial-fatigue loading. Subsequently, fatigue testing in CFRP cables subjected to the same fatigue loading as that applied to a CFRP rod was performed to evaluate the effect of bending anchoring. During the CFRP rod and CFRP cable fatigue tests, the multi-mode microscopic damage in CFRP rods was monitored using the AE technique.

### 2.1. Materials

In this study, the T700SC-12K carbon fibers from Toray Industries, Inc., Tokyo, Japan, and vinyl ester resins from Sinopec Group, Beijing, China, were used to fabricate a smooth-surfaced CFRP rod with a nominal diameter of 4 mm and a fiber-volume percentage of 66%. Their mechanical properties are presented in Table 1. The surface layer of the CFRP rod, which was resin-enriched in the anchoring section, was polished to reveal the carbon fiber and then cleaned using industrial alcohol (Ethanol 95%) to increase the bonding of the CFRP rod/cable specimen fabricated for fatigue testing.

### 2.2. Design and Fabrication of Bond-Friction Anchor for CFRP Rod Specimen

A novel bond-friction anchor, which prevents pull-out failure and compression-shear fracture of the CFRP rod inside the anchor under fatigue loading, is shown in Figure 1.

The proposed anchor for the fatigue test of the CFRP rod consists of a bonding section, a friction section, and an adapter connecting ring. The bonding and friction sections work together to ensure anchoring reliability and decrease the effect of anchoring. The friction section was fabricated by extruding a C45 steel-sleeve CFRP rod embedded using a die, which exceeds 80% of the CFRP rod fracture load. Thus, the friction section considerably reduces the expansion debonding rate in the bonding section by preventing slipping between the CFRP rod and the adhesive. On the other hand, the radial stress generated in the friction section can cause localized fiber bending, and the resulting concentrated premature fiber breakage often causes compression-shear fracture of the CFRP rod. To overcome these problems, a straight tube filled with potting compound (a mixture of modified high-temperature cured epoxy resin, cast stone powder, and steel grit) was used in the bonding section to reduce the tensile stress transferred to the friction section. This was useful to avoid the compression-shear fracture of the CFRP rod in accordance with the damage fracture criteria used in unidirectional FRP materials [16].

### 2.3. Design and Fabrication of CFRP Cables for Fatigue Testing

Fatigue-bearing capacity experiments of CFRP cables were performed on the bonding anchoring system. The fabricated CFRP cables for fatigue testing consisted of 37 CFRP rods (with a smooth surface and a diameter of 4 mm), and the gauge length of the cable was 1.0 m. The CFRP rods were diverged and arranged in an inner cone-shaped bonding anchor with a bonding length of 300 mm, and the maximal bending angle was set to 5.4°; the potting compound employed in the anchoring system was identical to that mentioned in Section 2.2. The fabricated CFRP cables are shown in Figure 2. More detailed design and fabrication information on the adopted bending anchoring system is available in the Refs. [17,18].

### 2.4. Fatigue Testing Setup for CFRP Rod and Cable

The maximum stress (*σ*_max_) and the stress amplitude (*σ*_a_), which are strongly related to the safety and service life of cables in infrastructure applications, are key parameters in cable design. For consistency with previously-obtained results from CFRP cable fatigue tests, in this study, the number of completed fatigue-loading cycles was set to 2 million, and *σ*_max_ and *σ*_a_ were selected as test variables. Based on theoretical and experimental studies on the fatigue properties of unidirectional FRP materials under axial fatigue loading, the following three typical degrees of damage caused to CFRP rods after applying 2-million fatigue-loading cycles:Mode I: Unaffected strength and stiffness;Mode II: Reduced strength and unaffected stiffness;Mode III: Fatigue failure (defined as a 30% reduction in stiffness in this study).

Consequently, the fatigue-loading conditions selected for this study must ensure that the fabricated CFRP rods exhibit the three degrees of damage mentioned above after applying fatigue loading. Table 2 summarizes the fatigue-loading conditions applied to the fabricated CFRP rods that were selected according to the preliminary fatigue testing (σ_ult_ = 2406 MPa is the strength with a 95% confidence level).

A 20-t electro-hydraulic servo fatigue testing machine was used to test the fatigue performance of the fabricated CFRP rod specimens. These specimens endured 4-Hz loading cycles until the fatigue stiffness decreased by 30% or the number of loading cycles reached 2 million.

Using a 100-t hydraulic-pulsed fatigue-testing machine, loading cycles at a 4 Hz frequency were applied to the CFRP cables by employing the force control mode, as shown in Figure 3.

The four CFRP cables (labeled as cable #1, cable #2, cable #3, and cable #4) were loaded according to the fatigue-loading conditions listed in Table 2. Equations (1) and (2) were used to obtain the nominal maximum loading force (*F_max_*) and the minimum loading force (*F*_min_) for each fatigue-loading condition.
*F*_max_ = *σ*_max_ × *S* × *n*(1)
*F*_min_ = (*σ*_max_ − *σ*_a_) × *S* × *n*(2)
where *S* is the cross-sectional area of a CFRP rod, and *n* is the number of CFRP rods in the cable. The CFRP cables were loaded until the fatigue stiffness decreased to 78% of its initial value (equivalent to the fracture of six CFRP rods) or the number of loading cycles reached 2 million.

In addition, for the analysis of the macroscopic damage progression during fatigue loading, two high-precision cameras were installed to monitor the damage of CFRP rods in the transition zone between the anchoring section and the free section of the cable.

### 2.5. AE Setup for Monitoring the CFRP Rods and Cables

An electromagnetic interference (EMI)-shielded lead zirconate titanate (PZT) patch (Baoding Yitian Ultrasonic Technology co., LTD, Baoding, Hebei, China) [17] was used as the AE sensor. During AE monitoring of the CFRP rods, the PZT patch was attached to the CFRP rod behind the friction section (no strain during the fatigue loading), as shown in Figure 4a. Thus, the AE monitoring results could reflect the damage caused to the CFRP rod. Moreover, during AE monitoring of the fatigue damage caused to the CFRP cables, the PZT patch was attached to the anchoring cup of the CFRP cable to obtain the AE signals, as shown in Figure 4b. The Physical Acoustic Corporation (Princeton, NJ, USA) DiSP-4/PCI system with digital passband filters (10 kHz–2 MHz), which cover all frequency ranges related to the damage patterns in CFRP materials, was employed. The system preamplifier was set to a 40 dB gain, and the acquisition trigger threshold was set to 42–45 dB (for the CFRP rods) and to 52–55 dB (for the CFRP cables). This arrangement could filter out any EMI generated by the fatigue testing machines at a zero-fatigue load.

## 3. Results and Discussion

### 3.1. Fatigue Resistance of the CFRP Rod under the Axial-Fatigue Loading

The fatigue stiffness of the CFRR rod specimens subjected to FLC-I, FLC-II, and FLC-III for two-million loading cycles was found to be stable. Furthermore, no macroscopic damage was found in the free section of the CFRP specimens. However, the fatigue performance of the specimens subjected to FLC-IV was quite different. All specimens failed, with the fatigue life ranging from 0.35 to 1.15 million loading cycles. The test results are summarized in Figure 5. Figure 5 shows that after two-million fatigue cycles, the residual strength retention rate of the CFRP rod was as high as 95.1% and 76.7% under stress amplitudes of 500 MPa and 600 MPa, indicating excellent fatigue resistance.

Figure 6 shows the macroscopic fatigue-damage progression of the CFRP rod under FLC-IV. As shown in Figure 6a, the initialization of macroscopic fatigue damage concentrated in the free section of the CFRP rod, causing fiber splitting (shown in white arrows) and fiber breakage (shown in red arrows) [29,30]. Figure 6b shows the fatigue failure of the CFRP specimens (with a 30% loss in stiffness). It was found that all of the fatigue damage distributed in the free section of the CFRP specimen and the fatigue-failed CFRP specimens still exhibited a certain amount of residual strength. Consequently, the residual strength of all CFRP specimens was examined. The results showed that only the CFRP specimens under FLC-III and IV sustained significant residual-strength loss. Then, the two-parameter Weibull model was used to evaluate the probability distributions of the residual strength of the CFRP specimens under FLC-III and IV, as shown in Figure 7.

As shown in Figure 7, the large variation in the residual strength of the CFRP specimens under FLC-III (all specimens were loaded for two million cycles) obviously reveals the unstable fatigue resistance of the CFRP rods since the residual strength of the CFRP specimens is largely related to the degree of fatigue damage. On the other hand, for the same level of reduced fatigue stiffness, the variation in the residual strength of the CFRP specimens under FLC-IV is smaller than that under FLC-III, indicating no significant difference in the degree of damage caused to the CFRP specimens under FLC-IV. However, the degree of damage caused to these CFRP specimens was obtained after applying quite different loading cycles, which also reveals the unstable nature of the CFRP rod fatigue resistance.

### 3.2. Macroscopic Fatigue-Damage Modes and Damage Progression in CFRP Cables

In this section, the macroscopic fatigue-damage modes and the progression in CFRP cables under different fatigue-loading conditions were investigated via analysis of the replays of the videos recorded by cameras during the whole fatigue-loading process. The cycle numbers corresponding to the damage initialization and fracture of CFRP cables are summarized in Table 3.

Cable #1 sustained no fatigue stiffness loss during the entire fatigue-loading process, and no macroscopic damage was found among the CFRP rods in the free section of the cable after applying two-million loading cycles.

In the fatigue testing of CFRP cables, several macroscopic damage modes were observed, which could not result in the extreme fracture of cables. However, some minor damages can occur with the accumulation of damage, such as the local fatigue fracture of CFRP rods in the free section of the cable due to defects in the material, as shown in Figure 8a. On the other hand, the predominant damage modes that are closely related to cable fracture are the main concerns in this section. The predominant damage modes are the fiber splitting of a CFRP rod in the free section of cable and the compression-shear fracture of a CFRP rod in the anchoring section of cable #2.

In the early stage (cycle from the beginning to the 1/3 of fatigue life) and intermediate stage (cycle from the 1/3 to the 2/3 of fatigue life) of the fatigue loading, the observed predominant damage mode in cable #2 is fiber splitting, which is prompted by the complex stress exerted on the CFRP rod in the anchoring section. As shown in Figure 8b,c, this damage mode starts in the anchoring section and progressively spreads to the free section of the cable. It may eventually separate the CFRP rod into several parts as fiber splitting propagates. Notably, the fiber splitting induced by the anchoring section affects the stiffness of the CFRP rod and intensifies the nonuniform distribution of stress among the CFRP rods. As a result, the bending of the CFRP cable and resulting shear further affect premature fiber breakage in CFRP rods. During the late stage of the fatigue-loading process, an asymptomatic compression-shear fracture of the CFRP rods in the anchoring section was observed, as shown in Figure 8d. Furthermore, all CFRP rods fractured in compression-shear mode also suffer severe fiber splitting. Therefore, it is reasonable to conclude that the degree of fiber splitting correlates closely with the level of premature fiber breakage.

The fatigue-damage modes in cable #3 are comparable to those in cable #2, but higher amplitude led to the concentrated and stochastic occurrence of the compression-shear fracture of the CFRP rods throughout the entire fatigue-loading process due to the high degree of stochasticity in the progression of the premature fiber breakage.

The greater the maximum stress applied to cable #4, the greater the weakening of the CFRP rod’s shear resistance, which made compression-shear fractures more likely. For cable #4, the compression-shear fracture of the CFRP rods in the anchoring section was the only mode of macroscopic damage that occurred during the entire fatigue loading without the occurrence of fiber splitting. Moreover, the damage characteristically progressed as a “strength fracture” with a high degree of stochasticity, which was quite different from that observed in cables #2 and #3.

### 3.3. Spatial Distribution of Macroscopic Fatigue Damage Caused to CFRP Cables

To evaluate the effect of bending anchoring on CFRP cables, the spatial distribution of the macroscopic fatigue damage caused to CFRP rods in the free section of the cable was further investigated after applying fatigue loading. In this study, three damage levels were introduced to define the degree of damage caused to CFRP rods at the free section of the cables: undamaged (no macroscopic damage), minor damage (local fiber splitting without forming separated CFRP parts, as shown in Figure 8c), and fatigue fracture (separated CFRP parts or compression-shear fracture in the anchoring section).

No macroscopic fatigue damage was observed in cable #1. Therefore, six outer- and inner-layer CFRP rods in cable #1 (shown in red and yellow in Figure 9) were removed from the middle of the free section of the cable.

The residual strength of these CFRP rods was tested, and the results described by the two-parameter Weibull model are presented in Figure 10. It is clearly observed that the residual strength of CFRP rods was not severely compromised, indicating that the CFRP rods at the free section of the cable were not affected by damage progression from the anchoring section.

All CFRP rods at the free section of cables #2, #3, and #4 were examined, and their fatigue-damage cross-sectional distributions are shown in Figure 11 (minor damage is shown in yellow and fatigue failure is shown in red).

It can be observed that all minor damaged or fatigue-fractured CFRP rods are concentrated at the outer layer of the CFRP rod bundle. This indicates that the fatigue resistance of the outer layer of the CFRP rod bundle (with the largest bending angle that suffers the most severe additional shear effect) significantly determines the fatigue-bearing capacity of bending-anchored CFRP cables.

### 3.4. Material-Scaled AE Monitoring of Fatigue Damage of CFRP Rod

As shown in Figure 4a, the obtained AE monitoring signals reveal the microscopic damages to the CFRP rod in the anchoring section. Therefore, the damage patterns in the AE signal can be identified using AE frequency analysis because of the inherent relationship between the AE peak frequency distribution and the microscopic damage patterns. Moreover, studies on the microscopic damage caused to FRP materials have shown that fiber-matrix debonding and fiber breakage are the main causes that affect the macroscopic properties of FRP materials (cited as critical damage patterns). Thus, the accumulated AE energy release due to critical damage patterns was adopted to assess the damage progression in the CFRP rod due to fatigue loading. In this section, the peak frequency of AE signals obtained under various fatigue loading conditions was analyzed using the fast Fourier transform to associate the damage patterns in the CFRP rod.

The AE frequency analysis indicated virtually no AE signal generation with a peak frequency above 150 kHz due to critical damage patterns under FLC-I and FLC-II. This means that according to the theoretical three-phase fatigue-damage model for unidirectional FRP materials under axial fatigue loading, the CFRP rod specimens subjected to FLC-I and FLC-II sustained no damage during the two-million fatigue-loading cycles.

Figure 12a shows the peak frequency distribution of the AE signals obtained from a typical CFRP specimen under FLC-III. A Few AE signals with a peak frequency above 300 kHz can be observed. The main critical damage pattern in the CFRP specimens under FLC-III is fiber-matrix debonding (with the AE peak frequency ranging from 150 kHz to 250 kHz). The normalized accumulated AE energy release due to critical damage patterns is shown in Figure 12b. It is observed that the fiber-matrix debonding curve exhibits a gradual linear increase. Based on the fatigue residual strength model, we can assume that the CFRP rods under FLC-III remain in the damage-stable expansion phase after applying two-million fatigue-loading cycles.

Figure 13 presents the AE monitoring of a typical CFRP specimen under FLC-IV. Figure 13a demonstrates that critical damage patterns released a significant amount more AE signals, and the AE hit in the time domain gradually increased during the fatigue-loading process. In Figure 13b, the accumulated AE energy release as a result of fiber breakage showed a gradual rise in the early stages of the fatigue-loading process and then a sharp rise. The trend of accumulated AE energy release due to fiber breakage corresponded with the progression of fiber breakage in the theoretical three-phase fatigue-damage model for the FRP under axial-fatigue loading.

Therefore, material-scaled AE monitoring revealed the damage status of CFRP rods post-fatigue loading (30% loss in fatigue stiffness with a certain residual bearing capacity) based on the characteristics of the microscopic damage progression described by AE monitoring. Moreover, the AE monitoring results also indicate that using the fabricated bond-friction anchor, premature fiber breakage in the anchoring section can be effectively restrained to a low level in the early stages of fatigue-loading progress, especially under FLC-IV. Therefore, the fatigue test results in Section 3.1 could represent the fatigue properties of CFRP rods under axial-fatigue loading.

### 3.5. Component-Scaled AE Monitoring on Fatigue Damage of CFRP Cable

The component-scaled AE monitoring focuses on the progressive fatigue-damage progression in cables #2 and #3. Figure 14 shows the variation of AE peak frequency distribution with the fatigue-loading cycles in cables #2 and #3, reflecting the brief damage progression in the anchoring section of the CFRP cable. It is worth noting that the curing residual stress in potting compounds and manufacturing errors in short CFRP cables (1.0 m) typically result in localized concentrations of damage in some CFRP rods. Due to the aforementioned causes, these effects result in localized damage concentrations in some CFRP rods at the initial fatigue-loading stage. During the fatigue-loading process, the curing residual stress in the potting compound and the degree of nonuniform stress distribution in the CFRP rod bundle will progressively decrease. Consequently, as shown in Figure 14, the AE signals due to fiber-matrix debonding are concentrated in the initial fatigue-loading stage. Moreover, it can be seen that the AE hits in cable #2 (especially with a peak frequency above 150 kHz) are distributed evenly, corresponding to the fiber splitting of the CFRP rod in the anchoring section. In contrast to cable #2, there are severe concentrations of AE hits in cable #3, corresponding to the occurrence of concentrated compression-shear fractures of CFRP rods.

Because the AE sensor was attached to the anchoring section cup (Figure 4b), it is important to note that the obtained AE signals were attenuated, which altered their frequency-domain characteristics. Therefore, AE frequency analysis could not be used to identify the damage patterns in the CFRP rods in the anchoring section. Based on the current research, the obtained AE signals were produced by the cracking of the adhesive and the damage caused to the CFRP rods, such as matrix cracking, fiber-matrix debonding, and fiber breakage. To determine the progress of the complex multi-mode damage observed in the anchoring section, the component-scaled AE monitoring results were analyzed using a cluster-based pattern recognition method. However, it was found that the number of AE signals caused by fiber breakage and fiber-matrix debonding is relatively low compared to the matrix cracking in CFRP rod and potting compound cracking (i.e., severely imbalanced class distributions in the AE signals). This study uses k-means clustering to identify damage patterns, which is highly sensitive to the degree of imbalanced class distributions in the AE signals. The k-mean clustering method is incapable of identifying damage patterns with only a small amount of AE signals (fiber breakage and fiber-matrix debonding), leading to unreliable clustering results. The subsequent procedures were, therefore, applied to the AE monitoring results. The AE primary peak frequency and logarithmic energy distributions in the frequency domain were employed as the AE feature vector to construct an AE dataset [17]. Then, several undersampling approaches were adopted to diminish the impact of the imbalanced class distribution in the AE dataset and improve the clustering quality. Clustering results using different undersampling approaches were evaluated using Davies-Bouldin values (i.e., the smaller the value, the better the separation effect), as shown in Figure 15.

The results of the AE clustering based on a random-undersampling approach achieved the optimal separation for both cable #2 and cable #3. The accumulated AE energy released due to the critical damage patterns in cables #2 and #3 is shown in Figure 16.

Based on the AE monitoring of the CFRP rod under axial tensile fatigue loading (i.e., FLC-II and FLC-III) and the observed compression-shear fracture of CFRP rods in cables #2 and #3, it was decided that the accumulated AE energy released due to fiber breakage showed the progression of the premature fiber breakage. Therefore, the accumulated AE energy released due to fiber breakage was regarded as a criterion for evaluating the compression-shear damage of the CFRP rods in the anchoring section.

The propagation of premature fiber failure monitored by AE is consistent with the macroscopic damage observed during fatigue loading. As depicted in Figure 16a, premature fiber breakage developed linearly in a slow and gradual manner within the 1.2-millionth cycle, and fiber splitting is the only damage mode observed in the early and middle fatigue-loading stages (initiated at the cycle range 312300–312800). Then, in AE monitoring, premature fiber breakage propagated rapidly, and the compression-shear fracture of CFRP rods occurred at a late stage (cycle ranges 1430420–1430450 and 1570160–1570187).

Figure 16b shows some abrupt increases in the progression of premature fiber breakage in AE monitoring as compared to cable #2. This indicated that the compression-shear fracture of the CFRP rod replaced fiber splitting as the predominant damage mode, which is consistent with the compression-shear fracture of CFRP rods in cable observed in this experiment (cycle range 410240–410250). This reflects the irregularity and high degree of stochasticity of the progression of premature fiber breakage under fatigue loading.

## 4. Conclusions

The fatigue resistance of bending-anchored CFRP cables was experimentally investigated to identify the macroscopic damage modes and progression. Furthermore, the AE technique was employed to monitor the multi-mode microscopic damage progression in CFRP rods in the anchoring section. The following conclusions can be drawn:Experimental results indicate that after the fatigue cycles of two million, the residual strength retention rate of the CFRP rod was as high as 95.1% and 76.7% under the stress amplitudes of 500 MPa and 600 MPa. Furthermore, the bending-anchored CFRP cable could withstand two million cycles of fatigue loading with a maximum stress of 0.4 *σ*_ult_ and an amplitude of 500 MPa without obvious fatigue damage; that is, the influence of the additional shear effect resulting from bending anchoring is negligible, demonstrating its superior fatigue resistance compared to steel cables. Nonetheless, under more severe fatigue-loading conditions compared to the fatigue properties of CFRP rod (Section 3.1), the bending-anchored CFRP cables suffer from a significant reduction in fatigue resistance due to the premature fiber failure caused by the additional shear in the bending anchoring section;The progressive fiber splitting of the CFRP rod in the free section of the cable and the stochastic compression-shear fractures of the CFRP rods in the anchoring system are the predominant macroscopic fatigue-damage modes observed under FLC-II and III. Under the more severe fatigue-loading conditions (FLC-IV), the stochastic compression-shear fracture of CFRP rods is the only macroscopic damage mode. Consequently, as the fatigue-loading level increases, the compression-shear fracture of the CFRP rods is likely to occur stochastically throughout the entire fatigue-loading process due to the high degree of stochasticity in the progression of premature fiber breakage;Minor damage or fatigue failure of CFRP rods in cable not only amplifies the fatigue loading on undamaged CFRP rods but also intensifies premature fiber breakage in CFRP rods due to the resulting shear from cable bending, likely leading to structural failure. The nonconvergent characteristics of the microscopic and macroscopic damages in the bending-anchored cable (Section 3.2 and 3.5) and the concentrated damage in the CFRP rod bundle (Section 3.3) support the conclusion that the robustness of fatigue resistance in bending-anchored CFRP cables is a critical issue that needs to be considered for the applications of CFRP cables in more severe fatigue environments;The spatial distribution of minor damage or fatigue fractures in the CFRP rods of CFRP cables shows a noticeable correlation with the extent of bending. Thus, the maximum bending angle becomes a critical issue in preventing the concentration of premature fiber breakage in CFRP rods under more severe fatigue environments. Therefore, the current maximal allowable bending angle in the literature that is determined according to the strength loss of a CFRP rod in bending anchoring mode is unable to ensure the prevention of the compression-shear fracture of a CFRP rod under severe fatigue conditions;Experimental observations have shown a considerable difference in fatigue resistance of the CFRP rod between axial tensile fatigue loading and complex stress in the bending anchoring system, primarily attributable to premature fiber breakage. Therefore, the curve trend of accumulated AE energy release due to fiber breakage obtained in the bending-anchored CFRP cable reflects premature breakage mode, making it a sensitive indicator to predict the compression-shear fracture of CFRP rods in the anchoring section.

## Figures and Tables

**Figure 1 polymers-15-02483-f001:**
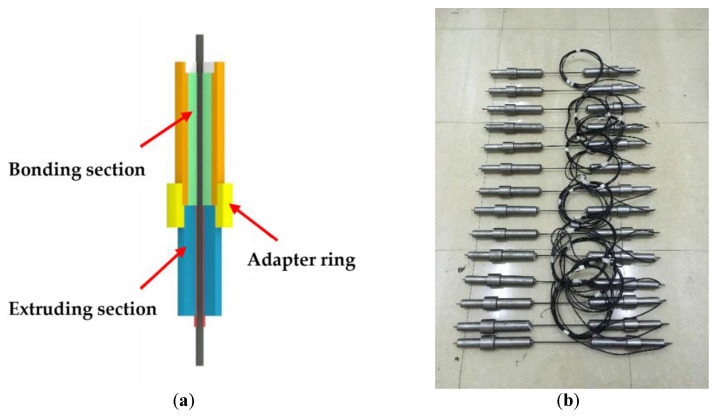
(**a**) Schematic of bond-friction anchor; (**b**) CFRP specimens fabricated for fatigue testing.

**Figure 2 polymers-15-02483-f002:**
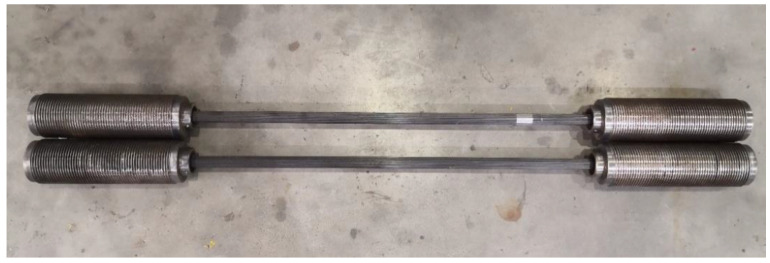
CFRP cables fabricated for fatigue testing.

**Figure 3 polymers-15-02483-f003:**
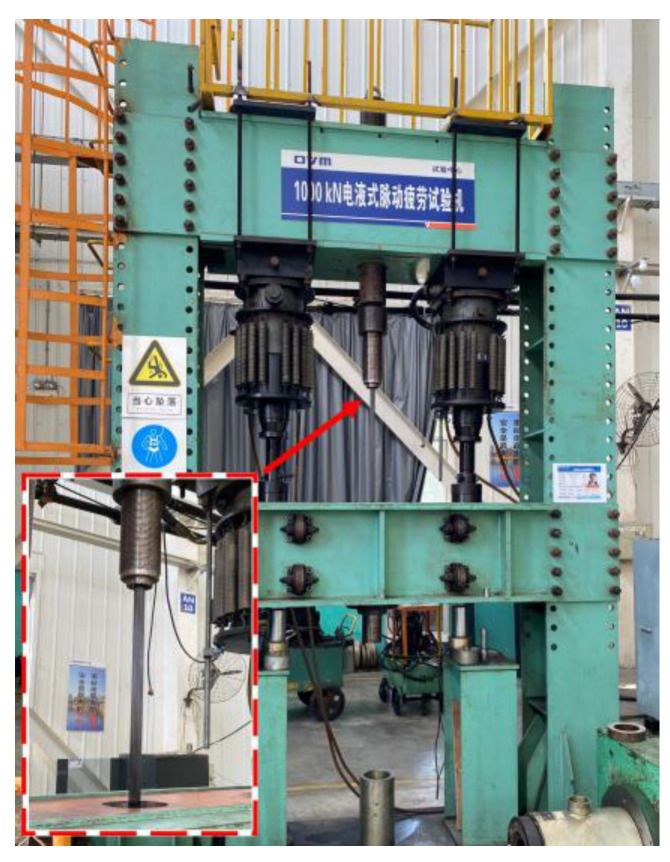
100-kN hydraulic-pulsation fatigue-testing machine.

**Figure 4 polymers-15-02483-f004:**
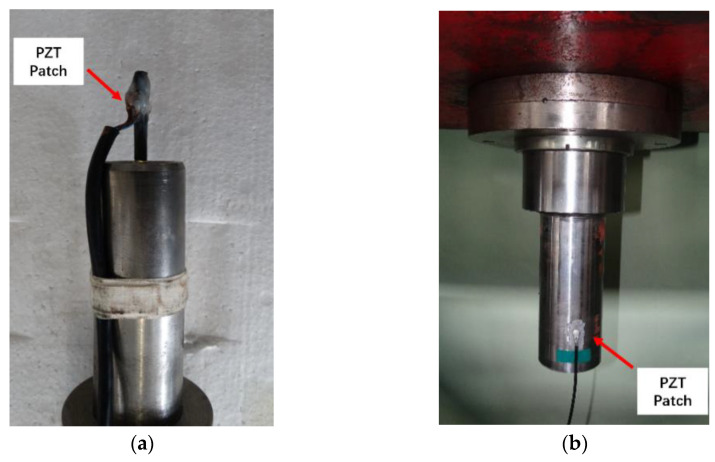
The PZT patch attached to the CFRP Rod (**a**) and CFRP Cable (**b**).

**Figure 5 polymers-15-02483-f005:**
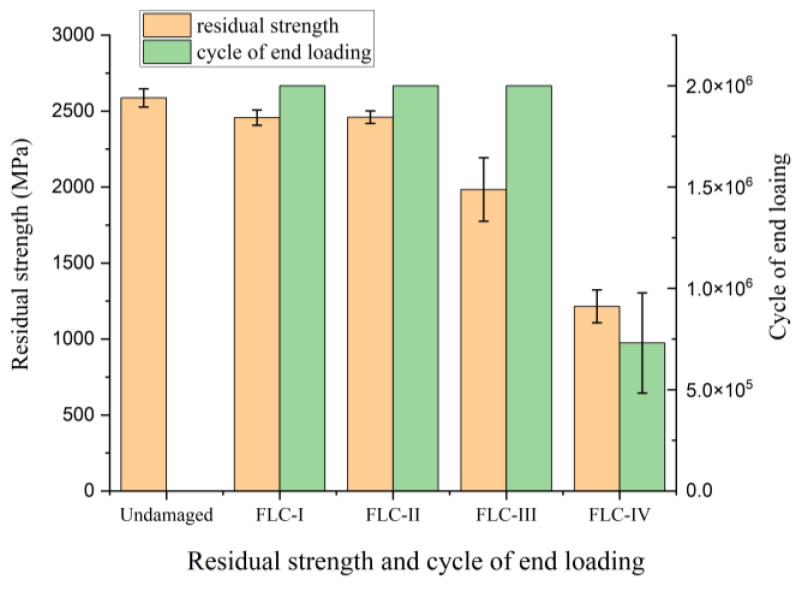
Fatigue testing results of CFRP specimens.

**Figure 6 polymers-15-02483-f006:**
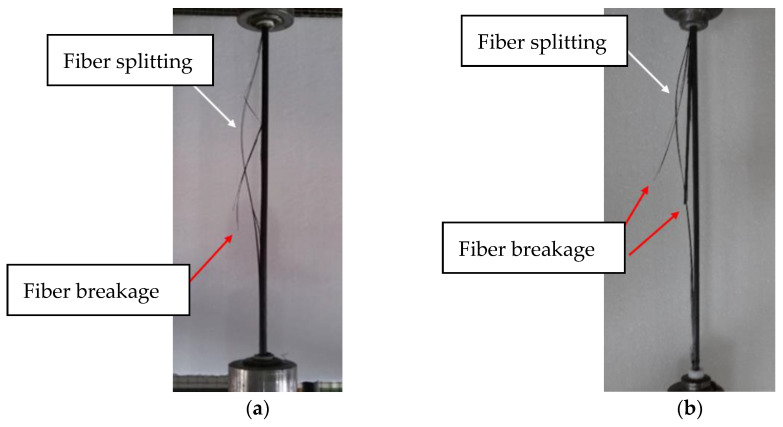
(**a**) Fatigue-damage initialization; (**b**) Fatigue failure in the CFRP rod under FLC-IV.

**Figure 7 polymers-15-02483-f007:**
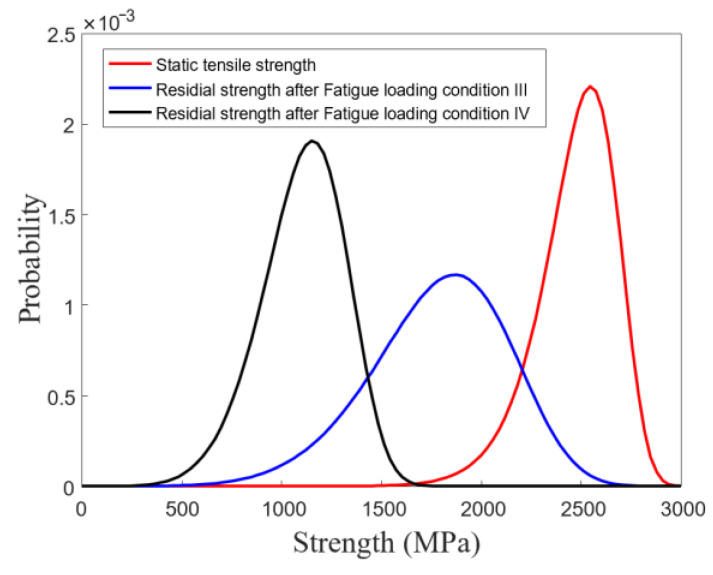
Probability distributions of the residual strength of CFRP rods.

**Figure 8 polymers-15-02483-f008:**
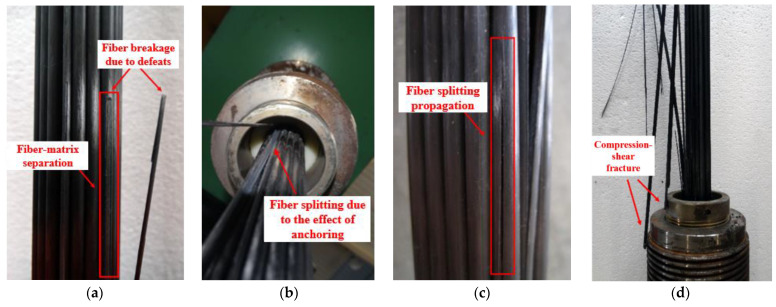
(**a**) Local fatigue fracture due to the defeats in CFRP rod; (**b**) Fiber splitting due to the effect of anchoring; (**c**) Expanding of fiber splitting; (**d**) Compression-shear fracture of CFRP rods in the anchoring section (cable #2).

**Figure 9 polymers-15-02483-f009:**
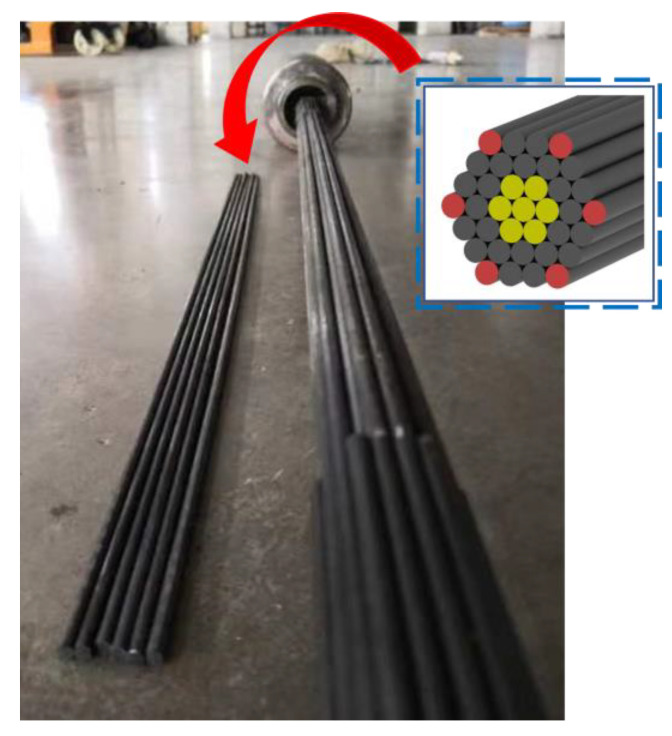
Six CFRP rods removed from the outer layer of CFRP rod bundle post-fatigue loading.

**Figure 10 polymers-15-02483-f010:**
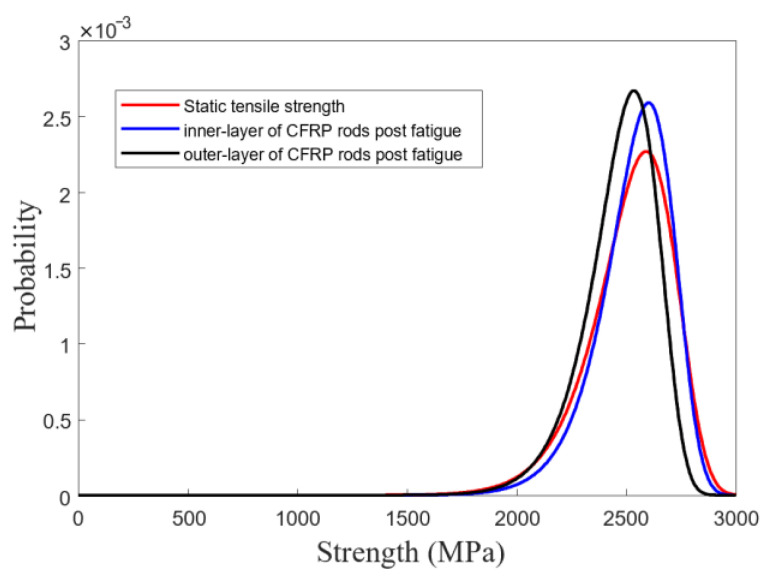
Residual strength distribution of CFRP rods in cable #1 after applying fatigue loading.

**Figure 11 polymers-15-02483-f011:**
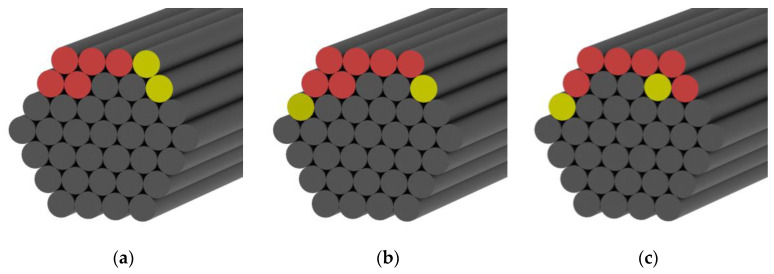
Spatial distribution of fatigue damage in cable #2 (**a**), in cable #3 (**b**), in cable #4 (**c**).

**Figure 12 polymers-15-02483-f012:**
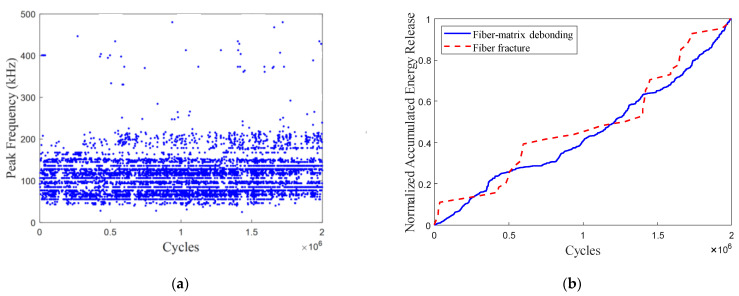
(**a**) AE peak frequency vs. fatigue-loading cycle under FLC-III. (**b**) Normalized accumulated AE energy release due to critical damage patterns vs. fatigue-loading cycle under FLC-III.

**Figure 13 polymers-15-02483-f013:**
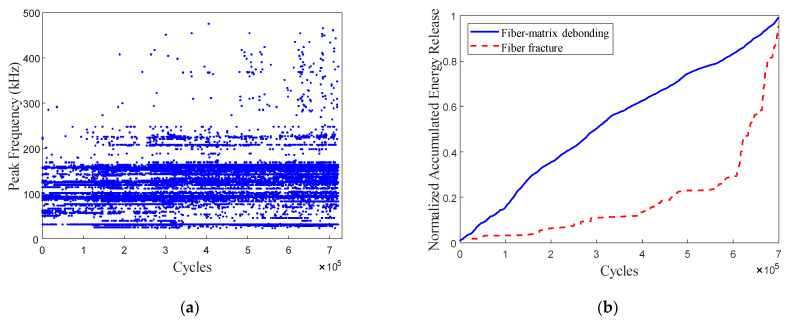
(**a**) AE peak frequency vs. fatigue-loading cycle under FLC-IV. (**b**) Normalized accumulated AE energy release due to critical damage patterns vs. fatigue-loading cycle under FLC-IV.

**Figure 14 polymers-15-02483-f014:**
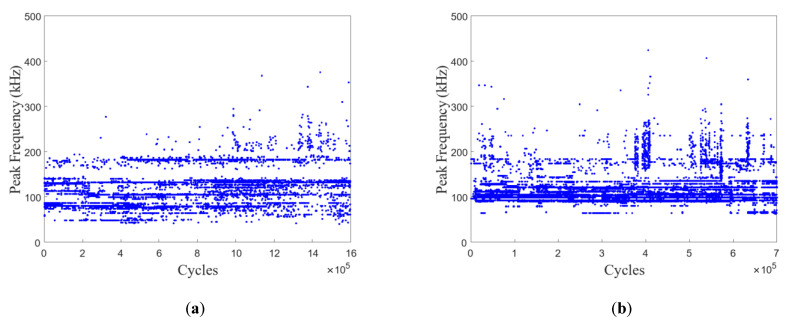
AE peak frequency distribution vs. fatigue-loading history in cable #2 (**a**) and in cable #3 (**b**).

**Figure 15 polymers-15-02483-f015:**
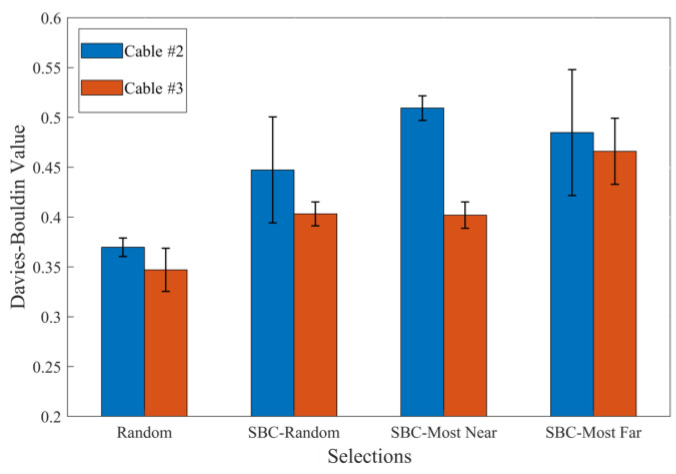
Davies-Bouldin value of the classifications of AE dataset (repeated 10 times).

**Figure 16 polymers-15-02483-f016:**
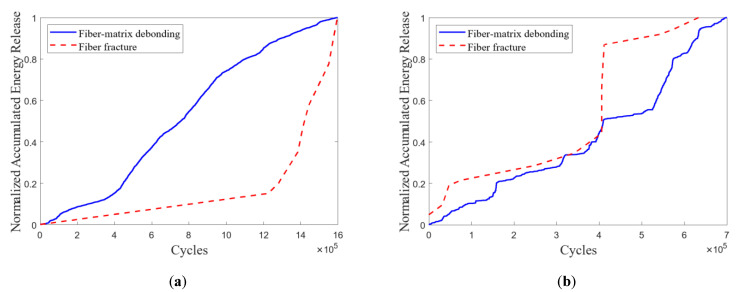
Normalized accumulated AE energy release vs. fatigue-loading cycle in cable #2 (**a**) and cable #3 (**b**).

**Table 1 polymers-15-02483-t001:** Mechanical properties of CFRP rod.

Material Type	Tensile Strength (MPa)	E-Modulus (GPa)	Elongation (%)
CFRP rod	2587	130	1.6

**Table 2 polymers-15-02483-t002:** Fatigue-loading conditions applied to the fabricated CFRP specimens.

Fatigue-Loading Condition	*σ*_max_ (MPa)	*σ*_a_ (MPa)	Loading Frequency (Hz)	Number of Test Specimens
FLC-I	0.4 *σ*_ult_	500	4	3
FLC-II	0.6 *σ*_ult_	500	4	3
FLC-III	0.6 *σ*_ult_	600	4	6
FLC-IV	0.8 *σ*_ult_	800	4	6

**Table 3 polymers-15-02483-t003:** Cycle of macroscopic fatigue damage in CFRP cables.

	Initialization of Fiber Splitting	Compression-Shear Fracture of CFRP Rods	Terminated Fatigue Loading
Cable #1	-	-	2 million
Cable #2	743,100–743,800	1,430,420–1,430,450; 1,570,160–1,570,187	1,570,187
Cable #3	312,300–312,800	410,240–410,250	680,158
Cable #4	-	35–40; 4585–4596; 7310–7324	7324

## Data Availability

The data presented in this study are available on request from the corresponding author.

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
