# Peer review of "Experimental Investigation of Fatigue Capacity of Bending-Anchored CFRP Cables"

_polymers, 2023, doi:10.3390/polym15112483_

Round 1

Reviewer 1 Report

We add the review results in the attachment

Author Response

Dear reviewer,

Thank you for your comments. Please find the attachment for relevant responses to the comments.

Reviewer 2 Report

In this study, the change in fatigue characteristic and fracture of CFRP rods in bend-clamped CFRP cables under severe fatigue loading conditions, were investigated to determine the adverse effects of the bending fixation system. Acoustic emission techniques were also used to monitor the multimodal microdamage progression of CFRP rods in a bending fixation system.

This paper analyzed the fatigue life of CFRP in great detail and precision and drew a logical conclusion. The paper is well written and of sufficient quality to be published in this journal.

Author Response

Dear reviewer,

Thank you very much for your positive and valuable comments.

Reviewer 3 Report

The paper presents an experimental investigation of the fatigue capacity of bond-friction anchored CFRP cables. A novel bond-friction anchor was used for the tests. The microscopic damage progression was monitored by acoustic emission and a total of 18 CFRP cables were tested for 4 fatigue load conditions. The failure mode was a combination of compression and shear, causing longitudinal splitting of the rods.

The paper presents an interesting topic and the testing procedure seems to be adequately designed and conducted. However, there are some issues regarding the presentation and writing- style of the paper that require clarification or improvement.

Do you really need to specify “fatigue-bearing capacity” in the title? In my opinion it is better to simplify and just refer to it as “fatigue capacity”.

The first sentence in the introduction needs to be rephrased.

In line 122 you write “In this study, the fatigue-bearing capacity of bending-anchored CFRP cable  under several specific fatigue loading conditions were evaluated experimentally to investigate the microscopic and macroscopic damage initialization, expansion, and fracture of CFRP rods in cable.” However, you never mention the aim of the paper and it would perhaps be better to rephrase this sentence as an aim.

In overall, the introduction needs to be extended and improved. Most references are quite old and the introduction does not provide a wide enough base for the readers as it looks now.

In the materials and methods, you need to further explain the anchor and which type of adhesives you used. Also you mentioned in some section that the diameter of the CFRP rod was 4 mm and in some other section you mentioned diameters up to 37 mm. This needs to be clarified to avoid confusion.  You are also mixing the term rods and cables, so you need to clarify what you are studying and perhaps distinguish the differences to the readers.

The bullet points in section 2.4 makes it a bit confusing. As I understand, all three points are for 2M cycles, which you mention in the text before. So therefore you can maybe simplify as: Mode 1: Unaffected strength and stiffness; Mode 2: Reduced strength, unaffected stiffness; Mode 3: Fatigue failure.

The results are relatively clear.

The first sentence in the first point of the conclusions need to be rephrased for clarity.

The paper does not seem to have been completely finalized since there are still parts that can be excluded, like section 6, supplementary materials, appendixes, etc.

There are a few language issues that needs to be improved, but the language is decent in general.

Author Response

(The authors gave the same response as above.)

Round 2

Reviewer 1 Report

After carefully check the revised manuscript, the authors are able to address all the issue that reviewers ask for clarification that occured at the previous version. We recommend accepted for the present form.

Reviewer 3 Report

Congratulations to the authors for writing an interesting paper. All my previous comments and suggestions have been answered and the proposed clarifications have been implemented. My recommendation is to accept the revised paper.